



GIS-models with fuzzy logic for Susceptibility Maps of debris flow using multiple types of parameters: A Case
Study in Pinggu District of Beijing, China
Yiwei Zhang[1], Jianping Chen[1,*], Qing Wing[1], Chun Tan[2,3], Yongchao Li[4,5,6], Xiaohui Sun[7], Yang Li[8]
1 College of Construction Engineering, Jilin University, Changchun 130026, China
2 China Water Northeastern Investigation, Design and Research Co., Ltd, Changchun, Jilin 130026, China
3 North China Power Engineering Co., Ltd. of China Power Engineering Consulting Group, Changchun, Jilin
130000, China
4 Key Laboratory of Shale Gas and Geoengineering, Institute of Geology and Geophysics, Chinese Academy of
Sciences, China.
5 University of Chinese Academy of Sciences.
6 Innovation Academy for Earth Science, Chinese Academy of Sciences, China.
7 Department of Earth Sciences and Engineering, Taiyuan University of Technology, Taiyuan 030024, China
8 Beijing institute of geological and prospecting engineering, Beijing 100020, China
* Corresponding author. Tel.:+86 13843047952
* Email address: chenjp@jlu.edu.cn
**Abstract**
Debris flow is one of the main causes of life loss and infrastructure damage in mountainous areas, so these
hazards must be recognized in the early stage of land development planning. According to field investigation and
expert experience, a scientific and effective quantitative susceptibility assessment model was established in Pinggu
District of Beijing. This model is based on Geographic Information System (GIS), combining with grey relational
method, data-driven and fuzzy logic methods. The inherent influence factors, which are divided into two categories,
are selected in the model consistent with the system characteristics of debris flow gully and some new factors are
proposed. The results of the 17 models are verified by the results published by the authority, and validated by the
other two indexes as well as Area Under Curve (AUC). Through the comparison and analysis of the results, the
method to optimize is proposed, including reasonable application of field investigation and expert experience,
simplification of factors and scientific classification. Finally, the final optimal susceptibility map with full
discussion has the potential to help in determining regional-scale land use planning and debris flow hazard
mitigation for decision makers, with full use of insufficient data, scientific calculation, and reliable results. The
model has advantages in economically backward areas with insufficient data in mountainous areas.
Key words: debris flow; susceptibility assessment; fuzzy logic; model optimization; hazard mitigation


## 1 Introduction

Debris flows are processes of rapid transport of water and soil materials in mountain watersheds, with sudden and destructive outbreaks(Di et al. 2019). Some debris flows can often cause devastating disasters and huge losses(Zhang et al. 2021) and seriously threaten the lives and properties of the people in the mountains, the safety of major projects, and restrict social and economic development (Hu et al. 2011; Hungr et al. 2005; Iverson 1997; Takahashi 2014; Wu et al. 2019). Mass movements in Beijing range in scale from shallow slope failures and rockfalls to catastrophic rock avalanches frequently mobilize to form debris flows, threatening the ecological environment of the mountainous area (Zhong et al. 2004). Especially, in recent years, due to the superposition of extreme rainstorm weather and human engineering activities, debris flow events have increased gradually(Li et al. 2021b). Besides, as the capital of China, Beijing has strong influence and radiation at home and abroad, where geological disasters are widely concerned (Li et al. 2020a; Xie et al. 2004). With the deepening understanding of debris flow disaster and the updating of database, a new and more accurate evaluation is also very necessary. Therefore, it is of great significance to establishing accurate and scientific debris flow susceptibility map.

Through previous studies, it can be summarized that the current research on debris flow mainly focuses on the following aspects: study on mechanism of debris flow, study on early warning and prediction of debris flow, study on numerical simulation of debris flow and study on debris flow hazard analysis. Especially, studies on debris flow hazard analysis have raised the attention of the researchers as soon as it appears(Dong et al. 2009). Communicating information about debris flow hazard analysis is a crucial component of preparedness and hazard mitigation(Chiou et al. 2015). Susceptibility assessment, an important part of a hazard assessment of geological processes is more flexible(Li et al. 2021a). In the early days, the susceptibility assessment of debris flows was mainly qualitative research. In 1976, the United Nations commissioned the International Union of Engineering Geology to conduct a risk assessment of debris flows, which marked the beginning of research on the susceptibility assessment of debris flows as an important research direction for disaster prevention and prediction (Li et al. 2020b). Many methods and techniques (Li et al. 2020b; Wu et al. 2019) have been proposed to evaluate debris flow susceptibility assessment based on different qualitative and quantitative approaches and geo-environmental information (Liu and Wang 1995).

The economy in mountainous areas is often backward, we cannot supervise and verify every basin due to the limited funds. Surely, they are also wasteful and unnecessary. The debris flow susceptibility assessment can give decision makers a basis for rational allocation of resources, and determine which gullies should be focused on. In other words, the study plays a link role for other studies. Recently, with the development of mathematical theory, computer technology, the application of 3S, the susceptibility assessment of debris flows has been extensively and quantitatively studied(Li et al. 2020b). While due to the nonlinearity of debris flow system and the openness and complexity of geological environment, we realize that it is chaotic, with many factors affecting the system. Therefore, it is very difficult to find a unified and standard evaluation model. At present, when the information is insufficient, the field investigation and experience of experts are necessary basis. However, the experience is often subjective and needs a lot of professional experience accumulation. Therefore, it is very important to express the experience of experts objectively and easily understandably to serve decision makers. The application of fuzzy set

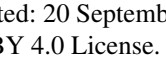
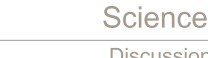
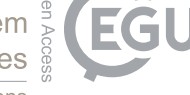

theory in GIS environments is effective for similar problems(Luo and Dimitrakopoulos 2003; Porwal et al. 2006).
According to the summary above, the primary object of my present study is to explore a geographic
information system(GIS)-based quantitative model based on expert experience and field investigation. And the
model is consistent with the system characteristics of debris flow gully and can also indicate the characteristics of
disaster chain and that the geomorphic evolution of basin rather than simple data fitting(Porwal et al. 2006).

**2 Study area**

The study area is located on the northeast of Beijing, China (Fig. 1), with a total area of 948.24 square
kilometers. The terrain of Pinggu is high in the northeast and low in the southwest. It is surrounded by mountains,
account for about two-thirds of the total area, on three sides in the southeast and north. The central and southern
parts are alluvial plains. The area , geologically, is the West extension of the famous Jixian section, whose bedrock
is mainly Middle and Late Proterozoic dolomite(Lü et al. 2017) .With Pinggu District of Beijing taken as the
research object, the following reasons are considered: First of all, geological hazards frequently influence human
economic activities, so political factors must be taken into account. And within the administrative region,
inconsistent decision-making can be effectively avoided. Next, the regional boundary is basically divided by ridge
line and stream line, and the regional geological environment is relatively uniform; Last but not the least, the
relationship between the precision of the base map and the size of the study area is also relatively reasonable.

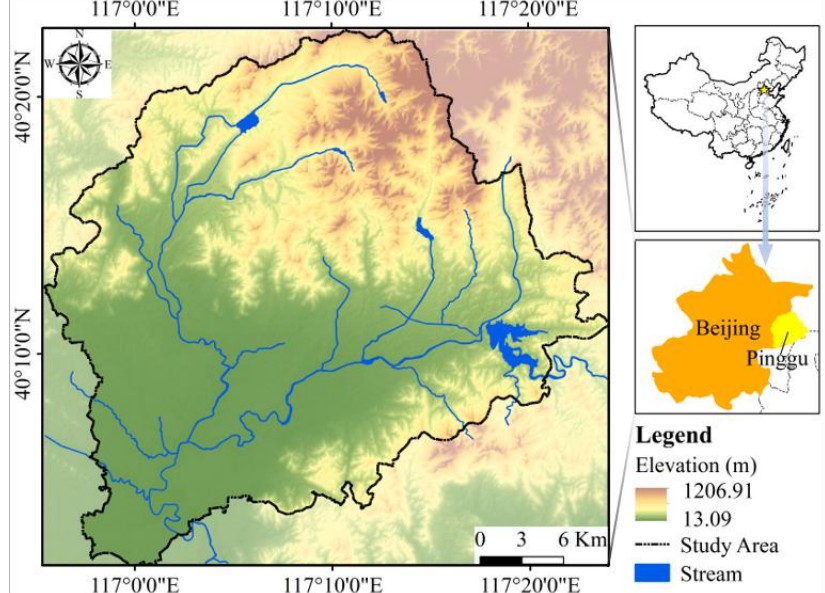

Fig. 1 Study area

**1. Data and Methodology**

In this study, the susceptibility assessment of debris flow hazard was based on the drainage basins unit. In a
debris flow susceptibility assessment model, hydro-logical response unit can fully represent the hydrological
process of hillside and will make the results more meaningful(Khan et al. 2013; Khan et al. 2016; Zou et al. 2019).




Therefore drainage networks were extracted from the ASTER-DEM by using the ArcGIS ArcHydro Toolbox and
regions without obvious watershed characteristics are directly deleted. Then for each drainage basin, 19 controlling
and triggering factors divided into two types were calculated. In addition, for these factors have different
characteristics, different methods are used to calculate the fuzzy membership for different type factors. Because the
field survey data are based on the watershed, it is scientific to make full use of qualitative understanding to
determine the weight of the parameters of watershed characteristics factors; while geology and geomorphology
factors are independent of watershed characteristics, it is suitable to use statistical methods to determine the
objective weight. Finally, the debris flow susceptibility index (DFSI) map was derived by overlaying the factor
thematic layers with fuzzy logic method. The workflow of debris flow susceptibility assessment is showed in Fig.2.
Throughout the modeling process, our primary assumption here are as follows: First, while local properties surely
affect the timing, size, and behavior of a mass movement, the dominant control on where they occur is the local
surface topography, as it in turn defines local slope and shallow subsurface flow convergence; Second, all the
evaluated basins have the possibility of debris flow; Thirdly, each evaluation factors should be available for all
basins, otherwise, it should be excluded; Finally, the model should also need to integrate the system characteristics
of debris flow disaster, the future development trend of climate change, and the social demand under the theoretical
background of the new era to carry out reasonable modeling.

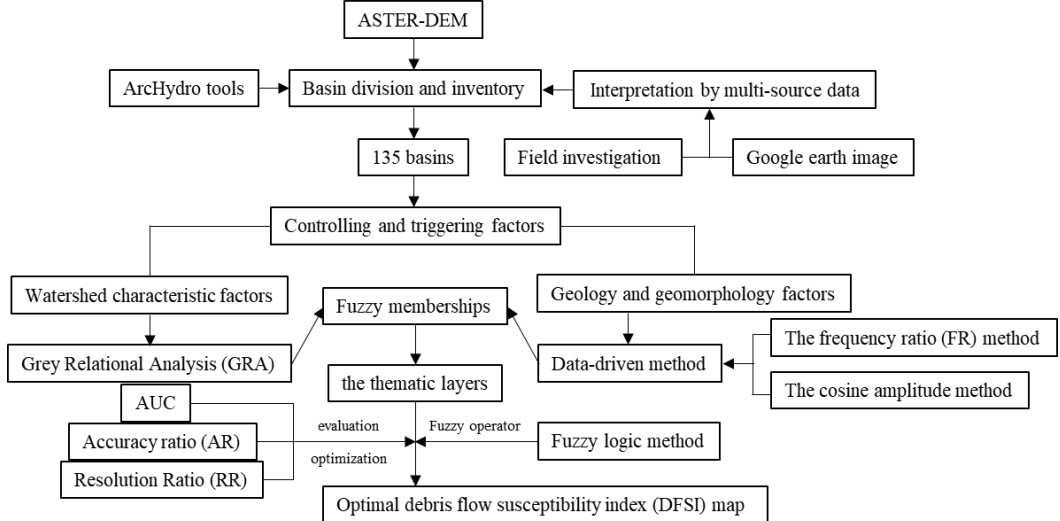

Fig.2 Workflow of debris flow susceptibility assessment
**3.1 Debris flow basin division and inventory**
There are many geological hazard points in mountainous area, so it is not realistic to monitor them completely
by professional team. According to the monitoring and preventing staff and the villagers, the detailed field
investigation (Fig.3) for the evidence collection of debris flows will be carried out at the reported disaster point,
aiming at record the loose material, delineating the basin and exploring other important information of the debris
flow gullies. Moreover, field investigation is also very important for model modification. Then based on the
Hydrology module in ArcGIS 10.2, the research object can be determined. Compared with grid unit and slope unit,



hydrological response unit for susceptibility of debris flow has greater advantages(Li et al. 2021b; Zou et al. 2019).
Finally, 135 basins are divided after removing the flat and irregular areas (Fig. 4), referring to the result of the field
investigation and the remote sensing image. In the 135 basins, 48 basins were investigated on field, accounting for

122    36%.

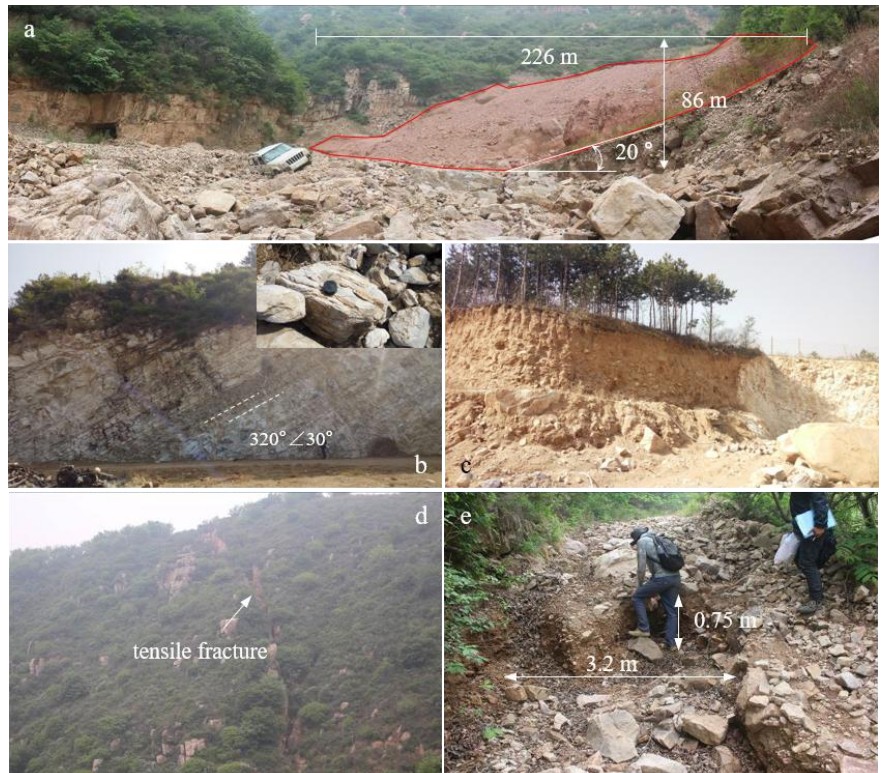


Fig.3 Field investigation photos. **a** Loose material; **b** Middle and Late Proterozoic dolomite; **c** colluvium deposit; **d**
Slope fracture; **e** Channel erosion phenomenon

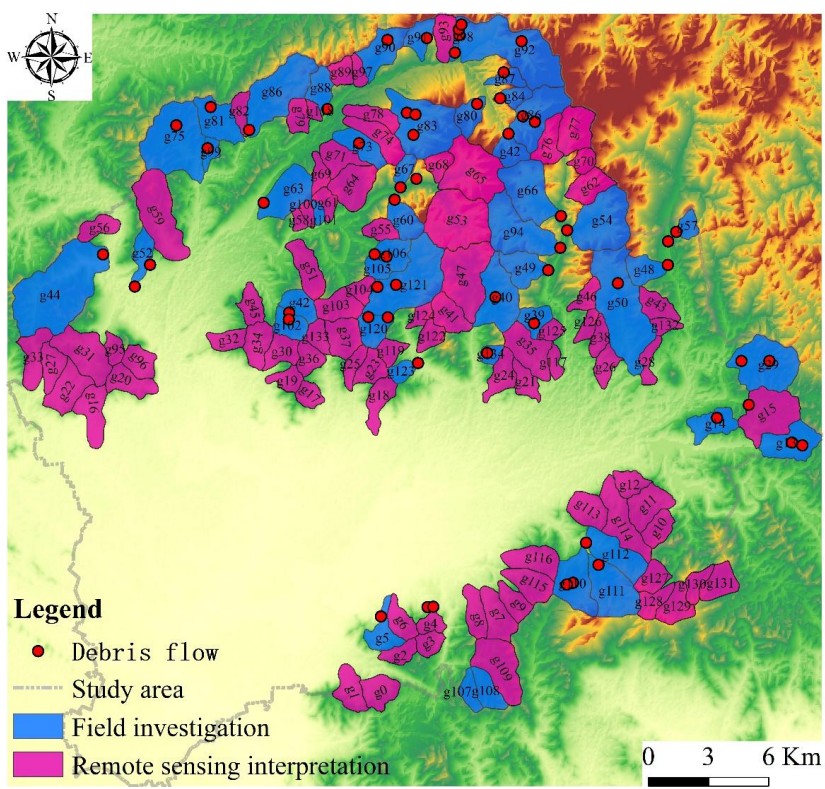

Fig. 4 Debris flow basin division and inventory.
Note: The data of debris flow points comes from Beijing Municipal Commission of Planning and Natural
Resources                                                                                                websites
(http://ghzrzyw.beijing.gov.cn/zhengwuxinxi/zxzt/dzzhfzzt/zzzhdcpg/202008/t20200807_1976436.html)
**3.2 Debris flow controlling and triggering factors**
The basic requirement for the assessment of debris flows is that at least some facros included are easily
obtainable, are meaningful for susceptibility assessment, and can be used for evaluating the need for passive or
active debris flow mitigation. According to previous studies, 19 factors are selected in this paper in this study. the
factors are divided into two types (Table 1) because of their different characteristics. Watershed characteristic
factors (Type A) can be directly quantified, once the basin is determined (Fig. 5). The influence of these parameters
is bounded by the watershed; Geology and geomorphology factors (Type B) factors need to be further processed,
even if the watershed is determined. The scope of these parameters is independent of the watershed boundary.
Besides, rainfall and total amount of loose material source are also very important influencing factors. But
according to the Beijing hydrological manual, the rainfall change in the study area is not obvious, so it is not
considered in my model. And the total amount of loose material source cannot be obtained for the watershed
without on-site investigation, so calculations are impossible. In fact, we indirectly consider the influence of natural
loose material source by evaluating geological conditions, but cannot consider the impact of human activities.



Table 1 Factors for susceptibility assessment

| Factors and Description | | | Significance | obtaining ways |
|---|---|---|---|---|
| Watershed characteristic factors (Type A) | $A_1$ | The planimetric (projected) area of the catchment | Geometric parameter; affecting the accumulative total volume of water and representing the potential magnitude(Cao et al. 2016; Chang and Chien 2007; Zhang et al. 2011) | derived from DEM |
| | $A_2$ | The curved surface area of the catchment | Real contact area between rainfall and basin | derived from DEM |
| | $A_3$ | The surface roughness of the catchment | Dimensionless parameters, reflecting the fragmentation degrees of the surface and the ground surface micro-topography. Wu et al. (2019) believe the factor can further reflects the ability of the earth to resist wind erosion. | Calculated by $A_3 = A_2 / A_1$ |
| | $A_4$ | The perimeter of catchment | Geometric parameter, controlling the boundaries of a watershed | derived from DEM |
| | $A_5$ | Form factor | Hydrologic parameter, related to the distribution of flow rate hydrograph(Chang and Chien 2007) | Calculated by $A_5 = \dfrac{A_4}{2\sqrt{\pi A_1}}$ |
| | $A_6$ | The curve length of the main channel | Importance for the travel distance of materials and affecting the potential of erosive agents to dislodge and transport materials(Gómez and Kavzoglu 2005) | derived from DEM |
| | $A_7$ | The straight length of the main channel | Geometric parameter, representing the change of material source in space | derived from DEM |
| | $A_8$ | Bending coefficient of the main channel | Affecting the discharge situation of debris flows(Li et al. 2020b; Zhang et al. 2013) | Calculated by $A_8 = A_6/A_7$ |
| | $A_9$ | The gradient of the main channel | Hydraulic gradient parameter, affecting water transport capacity | Calculated by $A_9 = A_{12}/A_6$ |
| | $A_{10}$ | Maximum elevation in the catchment | Affecting vegetation and bedrock exposure | derived from DEM |
| | $A_{11}$ | Minimum elevation in the catchment | Affecting vegetation and bedrock exposure slightly | derived from DEM |
| | $A_{12}$ | Maximum relative relief in the catchment | The higher the value of $A_{12}$ is, the large relative relief provides favorable terrain conditions for the initiation of the debris flow source. | Calculated by $A_{12} = A_{10} - A_{11}$ |
| | $A_{13}$ | Basin volume: the volume above the level of the minimum elevation in the basin | Representing the maximum material source that can be produced in an ideal state, loose material volume | derived from DEM |
| | $A_{14}$ | Drainage density | Representing the geological structure, lithology, and the degree of rock weathering comprehensively and affecting the range of lateral erosions and retrogressive(Cao et al. 2016; Zhang et al. 2011) | the ratio of the total length of river network lines to $A_1$ |
| Geology and geomorphology factors (Type B) | $B_1$ | Lithology | Affecting the rock mass shear strength and permeability (Donati and Turrini 2002) | derived from 1:50,000 numerical geological maps |
| | $B_2$ | Proximity to faults | correlated with slope failures by generally reducing the strength of the rock mass | derived from 1:50,000 |

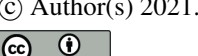


| | | | |
|---|---|---|---|
| | | (Dramis and Sorriso-Valvo 1994; Kellogg 2001; Korup 2004; Kritikos and Davies 2015). | numerical geological maps |
| $B_3$ | Slope (degrees) | correlated with the probability of landslide occurrence (Dai and Lee 2002; He and Beighley 2008; Lee and Choi 2004). The greater the slope, the greater the vertical component of gravity (Donati and Turrini 2002), and the higher frequency of slope failures (Lee and Sambath 2006; Lee and Talib 2005) | derived from DEM |
| $B_4$ | Slope aspect | affecting slope instability directly or indirectly, as a result of drying winds, sunlight, rainfall and vegetation (Dai and Lee 2002; Dai et al. 2001). | derived from DEM |
| $B_5$ | Curvature | Affecting slope stability. While Lee and Talib (2005) and Ohlmacher (2007) argue on how curvature affect slope stability. | derived from DEM |

Note: The geological maps are provided by Beijing institute of geological and prospecting engineering and the digital elevation model-(DEM) of study area are from SRTM-DEM with a solution. of 30 m (http://gdex. cr. usgs. gov/gdex/).

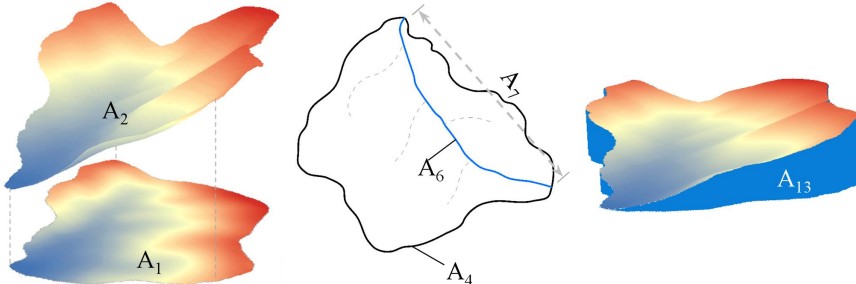

Fig. 5 Graphical illustration of some Type A factors. $A_1$ is the planimetric (projected) area of the catchment; $A_2$ is the curved surface area of the catchment; $A_4$ is the the perimeter of catchment; $A_6$ is the curve length of the main channel; $A_7$ is the straight length of the main channel; $A_{13}$ is basin volume

### 3.3 Fuzzy logic in susceptibility modelling

Fuzzy set theory proposed by Zadeh (1965) is a effective method to express the concept of partial set membership degree. This concept is different from the classical binary (two-valued) logic by using fuzzy descriptions such as low, moderate, high, steep, favourable and close to (Kritikos and Davies 2015). In the theory of fuzzy sets, elements have different degrees of membership in the interval [0,1]. 1 represents complete membership, and 0 represents non membership. Ross (1995) showed that fuzzy systems are useful in two general situations (Kritikos and Davies 2015). The method is very consistent with the characteristics of debris flow system, whose predisposing factors are fuzzy in nature and mechanism is complex and not fully understood. Application of fuzzy logic method, the most critical step is to find the suitable fuzzy membership of the factor. And fuzzy membership degree is equivalent to the weight in expert scoring method, which is calculated by objective method rather than given subjectively.





### 3.4 fuzzy memberships

#### 3.4.1 Grey Relational Analysis (GRA) in susceptibility modeling

GRA is proposed by Deng (1982) and it is an important part of grey system theory (Wang et al. 2014). Comparing with mathematical statistics methods which need lots of sample data, typical probability distribution and large calculation, GRA is applicable to small sample size and whether the data is regular or not. There will be no inconsistency between qualitative analysis and quantitative analysis (Deng 1988). Besides it is to excogitate the leading and potential factors that affect the development of the system, and quantitatively describe the development and change trend of the system by studying whether the relative change trend of the grey factor variables with complex relationship is consistent in the process of system development and evolution (Liu et al. 2004). Thus, grey correlation analysis is introduced to quantify the correlation between each factor and the evaluation results according to field investigation expert experience. First, the procedure of GRA is to translate the performance of every alternative into a comparability sequence (Kuo et al. 2008; Lin and Lin 2002; Wei et al. 2017). Therefore, according to technical standard, "Specification of geological investigation for debris flow stabilization (DZ/T0220-2006)", published by the China Ministry of Lands and Resources, the preliminary assessment results of debris flow susceptibility are obtained, which are used as the reference sequence of grey relation method (Table 2). Second, the grey correlation coefficient of all A factors is calculated by Eq. (1). Finally, the average grey relational coefficient (the correlation degree) is calculated by Eq. (2) as the fuzzy memberships (Table 3).

$$\xi_i(k) = \frac{\min_i \min_k |x_0(k)-x_i(k)| + 0.5 \max_i \max_k |x_0(k)-x_i(k)|}{|x_0(k)-x_i(k)| + 0.5 \min_i \min_k |x_0(k)-x_i(k)|} \tag{1}$$

Where $\xi_i(k)$ is the grey relational coefficient, i=1, 2, ..., n are the number i type A factors, k=1, 2, ..., n are the numbers of basin, $x_0(k)$ is the reference sequence (ideal target sequence), $x_i(k)$ is the number i type A factor sequence

$$r_i = \frac{1}{N}\sum_{i=1}^{n} \xi_i(k) \tag{2}$$

Where $r_i$ is the correlation degree in the range (0,1). N is the total number of basins in Table 2

Table 2 Quantitative evaluation grade standard table for Debris flow susceptibility

| gully | g5 | g13 | g14 | g29 | g39 | g40 | g42 | g44 | g48 | g49 | g50 | g52 | g54 |
|-------|-----|-----|-----|-----|-----|-----|-----|-----|-----|-----|------|------|------|
| score | 59 | 54 | 50 | 63 | 61 | 66 | 55 | 65 | 78 | 69 | 85 | 46 | 70 |
| gully | g57 | g60 | g63 | g66 | g67 | g72 | g73 | g75 | g80 | g81 | g83 | g84 | g85 |
| score | 56 | 63 | 58 | 73 | 62 | 84 | 62 | 67 | 84 | 69 | 80 | 75 | 86 |
| gully | g86 | g87 | g88 | g90 | g91 | g92 | g94 | g98 | g99 | g101 | g102 | g105 | g106 |
| score | 73 | 84 | 60 | 70 | 80 | 84 | 71 | 78 | 61 | 65 | 67 | 65 | 70 |
| gully | g107 | g108 | g110 | g111 | g112 | g120 | g121 | g123 | g134 | - | - | - | - |
| score | 45 | 45 | 69 | 69 | 74 | 62 | 63 | 73 | 56 | - | - | - | - |

Note: (130≥score ≥116, VH) , (115≥score ≥87, M) , (86≥score ≥44, L) , (43≥score ≥15, N)

VH=very high susceptibility, M=moderate susceptibility, L=low susceptibility, N= Non-debris flow

Table 3 The fuzzy memberships of type A factors

| Factor | A₁ | A₂ | A₃ | A₄ | A₅ | A₆ | A₇ |
|--------|----|----|----|----|----|----|----|
| Fuzzy membership | 0.77 | 0.77 | 0.63 | 0.6 | 0.54 | 0.55 | 0.67 |
| Factor | A₈ | A₉ | A₁₀ | A₁₁ | A₁₂ | A₁₃ | A₁₄ |
| Fuzzy membership | 0.71 | 0.55 | 0.55 | 0.59 | 0.61 | 0.79 | 0.54 |

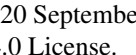



It can be seen from the results that the occurrence of debris flow is highly correlated with basin volume, basin area
and main gully bending coefficient with fuzzy membership above 0.7 in Beijing area. In the case of sufficient
rainfall, the basin directly determines the total amount of catchment, and the bending coefficient reflects the
replenishment of the source along the river. The basin volume is closely related to the number of supplementary
sources. Therefore, it is necessary to do well in rainfall monitoring and early warning in large watersheds, check for
loose matter accumulation in river basins before rainy season, and pay attention to slope protection of basin with
large volume potential energy for the purpose of disaster prevention and reduction.
**3.4.2 Data-driven method in susceptibility modeling**
Without regard to the influence of human activities, landslide is one of the main fixed sources of debris flow in
mountainous area. Shallow landslides are one of the most common categories of landslides. They frequently
involve large areas and different soils in various climatic zones (Benda and Dunne 1987; Borrelli et al. 2014; Selby
1982). Great debris flows may result from numerous, small slope failures that subsequently coalesce (Fairchild
1987; Roeloffs 1996), from flow enlargement due to incorporation of bed and bank debris (Bovis and Dagg 1992;
Pierson et al. 1990), or from large, individual landslides that mobilize partially or almost totally (Iverson et al. 1997;
Vallance and Scott 1997). Debris flows may also scour steep channels to bedrock and accelerate sediment delivery
to downstream, lower-gradient channels. The spatial and temporal distribution of shallow landslides are important
controls on landscape evolution and a major component of both natural and management-related disturbance
regimes in mountain drainage basins (Benda 1987; Crozier et al. 1990; Dietrich et al. 1986; Tsukamoto et al. 1982).
Therefore, the landslide susceptibility assessment methods can be used for reference to debris flow susceptibility
assessment.
For type B factors which cannot be characterized by a specific number, the frequency ratio (FR) method and
the cosine amplitude method can be used to derived their fuzzy memberships. The FR ratio defined as Eq. (3).
Considering the fuzzy membership must be in the interval [0,1], the FR values of the different categories are
normalized by the largest FR value (Lee 2006; Pradhan 2010; Pradhan 2011a; Pradhan 2011b) within the same type
factor ( Table 4) in order to derive the function.
$$FR = \frac{N_{(Di)}/N_{(Ci)}}{N_{(D)}/N_{(A)}} \tag{3}$$

where $N_{(Di)}$ is the number of debris flow pixels in the category i, $N_{(ci)}$ is the total number of pixels in the
category i, $N_{(D)}$ is total number of debris flow pixels in the study area, and $N_{(A)}$ is the total number of pixels in the
study area.

The cosine amplitude method (Ross 1995) is widely used (Ercanoglu and Gokceoglu 2004; Ercanoglu and
Temiz 2011; Kanungo et al. 2009; Kanungo et al. 2006) to establish relationships among elements of two or more
datasets (Kritikos and Davies 2015). Assuming that n is the number of data samples (categories of a factor used in
the analysis) represented as an array $X = \{x_1, x_2, ..., x_n\}$ and that each of its elements, $x_i$, is a vector of length m (i.e.
the size of the raster image) and can be expressed as $X = \{x_{i1}, x_{i2}, ..., x_{im}\}$, then each element of a relation $r_{ij}$ results
from a pairwise comparison of a factor category $x_i$ with a category of the debris flow distribution layer $x_j$ (debris
flow or non-debris flow). The memberships can be calculated by Eq. (4):



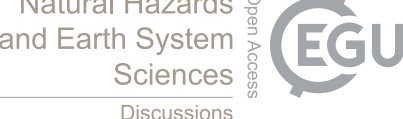
$$r_{ij} = \frac{\left|\sum_{k=1}^{m} x_{ik} x_{jk}\right|}{\sqrt{\left(\sum_{k=1}^{m} x_{ik}^2\right)\left(\sum_{k=1}^{m} x_{jk}^2\right)}} \tag{4}$$

Analogy with the study of Kanungo et al. (2006), we defined the $r_{ij}$ value for any given factor category as the
ratio of the total number of debris flow pixels in the category to the square root of the product of the total number
of pixels in that category and the total number of debris flow pixels in the area. Values of $r_{ij}$ close to 1 indicate
similarity whereas values close to 0 indicate dissimilarity between the two datasets (Kritikos and Davies 2015). In
order to use properly, every thematic layer must have the same pixel size.
Table 4 Factor categories and their fuzzy membership degrees

| Factor | Factor class | Number of pixels | Number of pixels classified as debris flows | Frequency ratio (FR) | Normalized frequency ratio | $r_{ij}$ | Compre-hensive ratio (FRR) |
|---|---|---|---|---|---|---|---|
| Lithology | Quanternary sediments-unconsolidatede clastic sediments | 7562017 | 48190 | 0.026 | 0.021 | 0.091 | 0.002 |
| | Coarse-grained sediments | 1148321 | 21741 | 0.076 | 0.063 | 0.061 | 0.004 |
| | Medium-grained sediments | 259619 | 12013 | 0.186 | 0.154 | 0.045 | 0.007 |
| | Fine-grained sediments | 754655 | 76380 | 0.407 | 0.337 | 0.114 | 0.038 |
| | High-grade metamorphics | 986435 | 154332 | 0.629 | 0.522 | 0.162 | 0.085 |
| | Granitoids | 725651 | 140936 | 0.781 | 0.648 | 0.155 | 0.100 |
| | Mafic extrusive | 75495 | 16398 | 0.873 | 0.724 | 0.053 | 0.038 |
| | Terrigenous clastic rock | 3289458 | 986495 | 1.205 | 1.000 | 0.41 | 0.410 |
| | Limestones | 8804379 | 1343754 | 0.614 | 0.509 | 0.478 | 0.243 |
| proximity to faults | <100 | 1057209 | 231016 | 0.878 | 1.000 | 0.198 | 0.198 |
| | 100-500 | 3778095 | 774566 | 0.824 | 0.938 | 0.363 | 0.341 |
| | 500-1000 | 3894600 | 716963 | 0.740 | 0.842 | 0.349 | 0.294 |
| | 1000-2000 | 5707265 | 760699 | 0.536 | 0.610 | 0.36 | 0.220 |
| | 2000-3000 | 2749240 | 246925 | 0.361 | 0.411 | 0.205 | 0.084 |
| | >3000 | 6421103 | 69382 | 0.043 | 0.049 | 0.109 | 0.005 |
| slope (degrees) | 0-5 | 9674508 | 153889 | 0.064 | 0.056 | 0.162 | 0.009 |
| | 5-10 | 2815606 | 383198 | 0.547 | 0.480 | 0.255 | 0.123 |
| | 10-15 | 2955913 | 521040 | 0.709 | 0.622 | 0.298 | 0.185 |
| | 15-20 | 2879704 | 570515 | 0.797 | 0.699 | 0.312 | 0.218 |
| | 20-25 | 2432724 | 498303 | 0.824 | 0.723 | 0.291 | 0.210 |
| | 25-30 | 1620325 | 350686 | 0.870 | 0.764 | 0.244 | 0.187 |
| | 30-35 | 837185 | 209574 | 1.007 | 0.883 | 0.189 | 0.167 |
| | 35-40 | 294141 | 82000 | 1.121 | 0.983 | 0.118 | 0.116 |
| | 40-45 | 77038 | 21133 | 1.103 | 0.968 | 0.06 | 0.058 |
| | >45 | 30091 | 8529 | 1.140 | 1.000 | 0.038 | 0.038 |
| Slope aspect | Flat | 380875 | 463 | 0.005 | 0.005 | 0.009 | 0.000 |
| | North | 2370048 | 296900 | 1.006 | 1.000 | 0.318 | 0.111 |
| | Northeast | 2193998 | 279917 | 0.513 | 0.510 | 0.218 | 0.092 |





|  | | | | | | | |
|---|---|---|---|---|---|---|---|
|  | East | 2873308 | 295555 | 0.414 | 0.411 | 0.224 | 0.111 |
|  | Southeast | 3122267 | 353489 | 0.455 | 0.453 | 0.245 | 0.108 |
|  | South | 3219111 | 354420 | 0.443 | 0.440 | 0.246 | 0.133 |
|  | Southwest | 3144353 | 400064 | 0.512 | 0.509 | 0.261 | 0.135 |
|  | West | 3525895 | 436381 | 0.498 | 0.495 | 0.273 | 0.140 |
|  | Northwest | 2787380 | 381679 | 0.551 | 0.547 | 0.255 | 0.318 |
|  | Concave | 490900 | 109157 | 0.893 | 1.000 | 0.136 | 0.136 |
|  | Lessconcave | 2037602 | 394583 | 0.778 | 0.871 | 0.259 | 0.226 |
| Curvature | Flat | 18364429 | 1769210 | 0.387 | 0.433 | 0.549 | 0.238 |
|  | Less convex | 2202019 | 416142 | 0.759 | 0.850 | 0.266 | 0.226 |
|  | Convex | 522285 | 112740 | 0.867 | 0.971 | 0.139 | 0.135 |


## 3.5 DFSI map


To derive the debris flow susceptibility index (DFSI) map by overlaying the factor thematic layers using fuzzy
logic method, the "fuzzified" factors represented by information layers in raster format with values ranging from 0
to 1 need to be combined. Compared with other four fuzzy operator, Fuzzy Gamma (Eq.6) is more suitable for the
research (Kritikos and Davies 2015). To determine the appropriate γ value, the results of different gamma values
were compared by the greatest distance (Kritikos and Davies 2015) between the average DFSI curves of the debris
flows locations and non-debris flows locations (For example, flat pixels)(Fig. 6). Finally, 0.9 is determined for the γ
value, because there is the greatest difference between debris flow and non-debris flows locations areas. In order to
illustrate the superiority of our model through comparison, seventeen results are calculated in ArcGIS (Fig. 7).
$$\mu_{(x)} = \left(1 - \prod_{i=1}^{n}(1 - \mu_i)\right)^{\gamma} * \left(\prod_{i=1}^{n}\mu_i\right)^{1-\gamma} \tag{5}$$

where $\mu_{(x)}$ is the combined membership value, $\mu_i$ is the fuzzy membership function for the ith map, i=1,2, …, n
are the numbers of thematic layers to be combined, and γ is a parameter in the range (0,1).


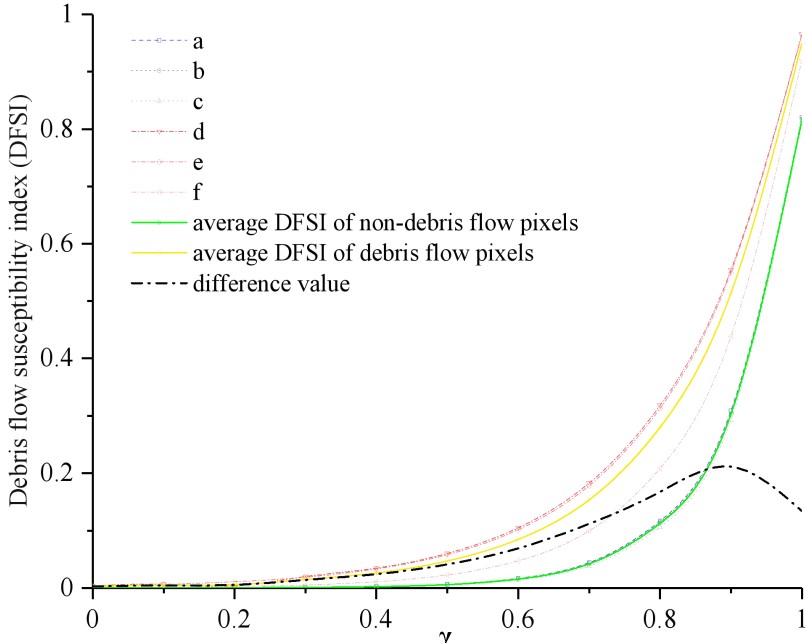

Fig. 6 Effect of γ value on Debris flow susceptibility index (DFSI). Curves d, e and f correspond to debris flow
pixels, and curves a, b and c correspond to non-debris flow area where a Debris flow is unlikely. According to
curve i, the maximum difference between the average DFSI values is observed for γ≈0.9
In order to find the optimal model, seventeen results were compared (Table 6). According to the distribution
map of potential geological hazard points and susceptibility map in Pinggu District published by Beijing Municipal
Commission of Planning and Natural Resources(BMCP&NR 2020), three indexes are used to verify the validity
and accuracy of the model.
The results of the model are independent of the model itself, so the predictive performance of the final map is
not just "the goodness of fit" of the data (Chung et al. 1995; Remondo et al. 2003). A relatively reliable technique
for quantitatively assessing how well a model is the construction of validation or success rate curves (Chung and
Fabbri 1999; Frattini et al. 2010; Remondo et al. 2003; Westen et al. 2003) based on a comparison between the
spatial distribution of debris flows and modelled debris flow susceptibility. The curves illustrate the debris flow
recorded in the area with respect to susceptibility values also expressed as cumulative percentages of the total area.
The area under the curve (AUC) defines the success rate (Marjanović et al. 2011). Generally, AUC values above 0.7
indicate model performance can be acceptable, while below 0.7, the performance is considered poor (Kritikos and
Davies 2015).
Although AUC is an effective evaluation method, the results is not comprehensive as mathematical features
for selecting the best measurement model because of insufficiency data for validation. In order to ensure the
objectivity of the results, we can only effectively use the recorded debris flow gully as positive, while the others as
negative. Thus, a two-category test is proposed to verify the model in this paper. First, the DFSI map of each model
are divided into two categories by Natural Breaks (Jenks) method (Fig. 7). Then the accuracy ratio (AR) is defined

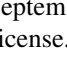


as the frequency of the number of debris flow both classified by model and simultaneously recorded in site to the
number of debris flow recorded in site. The Resolution Ratio (RR) is defined as the number of debris flow
classified by model and simultaneously recorded in site to the total number debris flow classified by the model (in
red color). Take $R_4$ for example, there are total 135 basin in the research area, but only 46 records of debris flows
(Fig.3). And in the results of two categories by Natural Breaks (Jenks) method, 20 basins are divided in to debris
flow, while there are only 14 debris flows among them. Then AR is calculated by dividing 14 into 46 and RR was
calculated by dividing 14 into 20.

The higher the two values, the better the susceptibility map. Finally, the performance of models (P value) can

be obtained by the Eq. (6). AUC values less than 0.6 are directly eliminated. Comparing the results of rest models,
the result of $R_{16}$ is optimal, and the results of DFSI map are in good agreement with those of field investigation (Fig.

8).

$$P = AUC + \sqrt{(AR * RR)} \tag{6}$$
Table 5 Predictive performance of different models

| Result and Description | | AUC | Two-category test | | Performance index (centesimal grade) |
|---|---|---|---|---|---|
| | | | Accuracy Ratio (AR) | Resolution Ratio (RR) | |
| A factors only or B factors only | $R_1$ — B factors with $r_{ij}$ | 0.460 | / | / | / |
| | $R_2$ — B factors with FR | 0.687 | / | / | / |
| | $R_3$ — B factors with FRR | 0.602 | / | / | / |
| | $R_4$ — All A factors | 0.786 | 0.304 | 0.700 | 83 |
| | $R_5$ — Selected A factors | 0.760 | 0.391 | 0.750 | 94 |
| All factors as a single thematic layer | $R_6$ — All A factors and B factors with $r_{ij}$ | 0.776 | 0.261 | 0.667 | 74 |
| | $R_7$ — All A factors and B factors with FR | 0.779 | 0.283 | 0.684 | 78 |
| | $R_8$ — All A factors and B factors with FRR | 0.753 | 0.326 | 0.600 | 76 |
| | $R_9$ — Selected A factors and B factors with $r_{ij}$ | 0.746 | 0.348 | 0.727 | 86 |
| | $R_{10}$ — Selected A factors B factors with FR | 0.761 | 0.348 | 0.727 | 87 |
| | $R_{11}$ — Selected A factors B factors with FRR | 0.740 | 0.348 | 0.727 | 85 |
| A factors combined into one thematic layers, B factor combined into another thematic layers | $R_{12}$ — All A factors and B factors with $r_{ij}$ | 0.708 | 0.5 | 0.511 | 82 |
| | $R_{13}$ — All A factors and B factors with FR | 0.753 | 0.848 | 0.394 | 99 |
| | $R_{14}$ — All A factors and B factors with FRR | 0.711 | 0.870 | 0.404 | 96 |
| | $R_{15}$ — Selected A factors and B factors with $r_{ij}$ | 0.726 | 0.348 | 0.667 | 80 |
| | $R_{16}$ — Selected A factors and B factors with FR | 0.768 | 0.739 | 0.442 | 100 |
| | $R_{17}$ — Selected A factors B factors with FRR | 0.740 | 0.457 | 0.600 | 88 |

Note: Selected A factors with fuzzy membership more than 0.6; FRR represents the product of FR and $r_{ij}$;
Performance index is normalized by the largest FR value


Fig.7 Results of two categories by Natural Breaks (Jenks) method





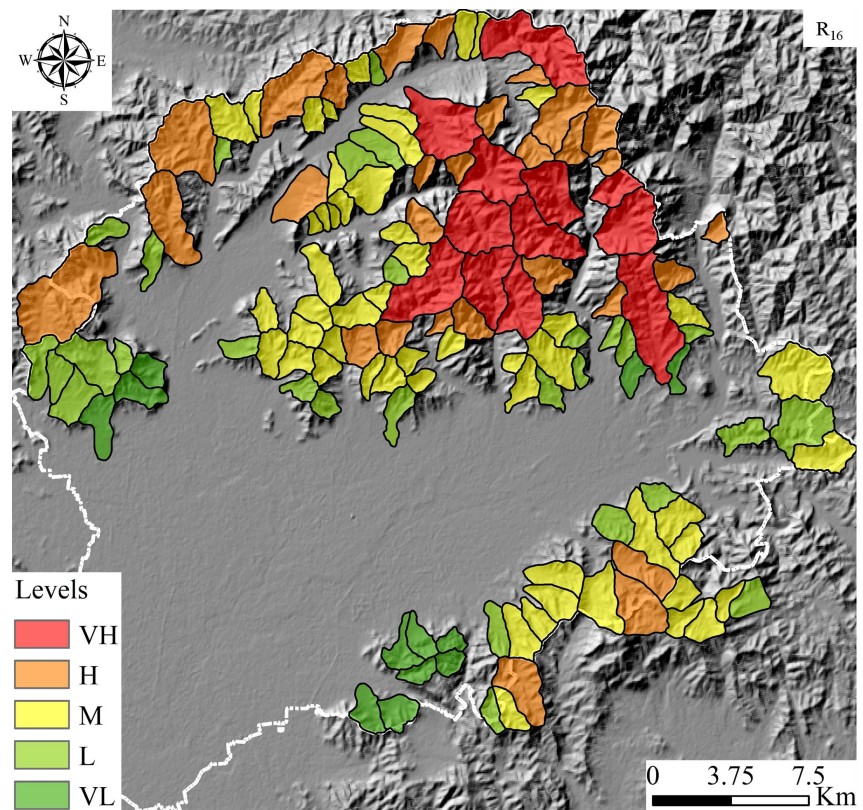

Fig 8 Debris flow susceptibility maps

**4 Results and Discussion**

According to the previous researches, 19 factors are selected. Although these factors cannot fully evaluate the character of a basin, it is necessary to consider that they are easily obtainable for each basin and can be obtained relatively accurately, ensuring that the model can be widely applied. Vegetation and rainfall factors are also very important, but there is little difference in vegetation and rainfall across the study area. Considering the background of global climate change, high temperature and extreme rainfall events will be increasing, which also makes them uncertain factor compared with factors compared. As for the factors describing debris flow magnitude, usually, several channels have the recorded data. Other factors that also influence the susceptibility of debris flow are usually difficult to obtain, including soil drainage, induration, thickness, conductivity, and strength properties; subsurface flow orientation; bedrock fracture flow; and root strength.

The predictive performance of the output debris flow susceptibility maps, obtained from seventeen different models, is verified by comparing with maps published by authority. By comparing the results, the following results are discussed:

First, comparing $R_1$, $R_2$, $R_3$, $R_4$ and $R_5$, it can be concluded that the model based on field investigation and expert experience is more effective than data- driven directly, when the sufficient information cannot be obtained. This is mainly because when the basin area reaches a certain size, it is no longer controlled by one or several



factors, but becomes a complex system. It is not only the factors that affect the system, but also the system will react on each factor. Geomorphic evolution is basically the result of the interaction of the endogenic and exogenic geological processes. A geological period can be regarded as the beginning of an endogenic geological processes to the next one. In the early stage of geological period, endogenic geological processes play a major role, and in the later relatively stable period, exogenic geological processes will play a more and more important role. In this large cycle, the basin continuously occurs a small cycle of accumulating and releasing energy, which leads to extremely complex system changes. In addition, there is a contradiction between the scale of geological evolution and the scale of engineering activities. So limited information can be obtained under these conditions that leads to the unreliability of data-driven evaluation. Therefore, in the current period, field investigation and expert experience are fundamental.

Second, by comparing $R_4$ and $R_5$, $R_6$ and $R_9$, $R_7$ and $R_{10}$, $R_8$ and $R_{11}$, $R_{12}$ and $R_{15}$, $R_{13}$ and $R_{16}$, $R_{14}$ and $R_{17}$, it can be concluded that the accuracy and resolution of the model can be improved by simplifying the factors, which will eliminate the weak correlation and independence factors. In practical application, even if the susceptibility map is obtained, the classification of the susceptibility degree is still a very difficult problem. Because everyone's subjective definition of "susceptibility degree" is different. By simplifying the factors, the main factors can be selected, which magnifies the differences between basins, so the boundaries between different susceptibility degrees are more obvious.

Third, by comparing $R_6$ and $R_{12}$, $R_7$ and $R_{13}$, $R_8$ and $R_{14}$, $R_9$ and $R_{15}$, $R_{10}$ and $R_{16}$, $R_{11}$ and $R_{17}$, it can be concluded that the model in which factors are classified into two types is better than the method in which all factors as a single thematic layer without classification. Because the factors categorized separately are more closely linked and has consistent influence on the system in mechanism. We can also infer that the non-linear combination characteristics between different types are stronger and scientific classification can improve the performance of the model.

Fourth, comparing $R_{12}$ and $R_{13}$, $R_{15}$ and $R_{16}$, it can be concluded that the frequency ratio method is better than the cosine amplitude method in the study. Different from the study of Kritikos et al. (2015), the watershed unit rather than the grid unit is used, which indicates that the former has a wide range of application, while the latter has a disadvantage of strict conditions.

Based on the results of the above four analyses, the most optimal model should have the features of being based on expert experience, using selected factors, classifying factors before using them, and using frequency ratio method. Then the model $R_{16}$ is selected according to the features, which is well in accordance with theoretical method performance score, and gets fine mutual verification.

In summary, the debris flow susceptibility assessment in this study follows the principles of scientific and practicality. First, classification of influencing factors follows the principles of scientific, which require the classification to be accurate and systematic. Then the same susceptibility degree can be classified into the same type reasonably. In order to correctly classify the factors, it is necessary to grasp the characteristics of the formation, movement and accumulation of debris flow. Therefore, the classification should comprehensively consider the development background (geology, geomorphology, climate, hydrology, soil, vegetation, human activities and other factors). The practical principle refers to that the study should not only fully obtain scientific and accurate results,



but also make the professional results understood by decision makers. The relative simplicity of the model with
data easy to obtain is attractive, which can also provide necessary information for debris flow mitigation and land
utilization. Although the susceptibility grade and susceptibility value of each watershed is obtained, the results are
relatively effective in this study area. The purpose is to distinguish the difference of each channel for
decision-making to work out pertinence measure. Once separated from this study area, the comparison with other
regions in value will lose its practical significance. In addition, with the development of technology and theory, we
should replace some traditional factors which are not easy to quantify with more precise quantitative factors to
improve the efficiency and accuracy of evaluation, such as surface roughness instead of drainage density. Last,
nonlinear methods is consistent with the nonlinear characteristics of debris flow system.
**5 Conclusion**
In the present study, a new combination model for debris-flow susceptibility based on GIS was developed in
Pinggu, the eastern of Beijing. The objective and motivation of this study is to demonstrate a simple, extensible,
and convenient analytical model for the debris flow prediction. Three methods are selected in the model with their
own advantages. GRA has great advantages in the case of less samples, data-driven method is mainly used to
reduce subjectivity and fuzzy logic is fitted to solve nonlinear problems with fuzzy classification. The output debris
flow susceptibility maps obtained from the optimal models demonstrated satisfactory performance predicting
approximately 50 % of the debris flow gully with the relative higher susceptibility values corresponding to
AUC≥0.7. Considering that the data used for verification is only the recorded debris flow points rather than all
debris flow records in the area, its accuracy should be higher. The predictive performance of the susceptibility maps
and the spatial correlation of debris flow gully with H and VH susceptibility with recorded debris flow illustrate
that the assessment at regional scale using the proposed method is feasible. Compared with the previous results
based on grid units in this area, the evaluation results are basically the same, but they are more targeted for debris
flow disasters for decision makers{Li, 2020 #278}.Besides, considering that the meaning of the used factors is
clear and the data easy to obtain, these conditions mentioned enable the model to be widely applied.
Preliminary research indicates that: First of all, the relatively ideal evaluation results are obtained by
combining the landslide susceptibility analysis method with the debris flow. It reveals a systematic idea and disaster
chain phenomenon. Further more, we should pay more attention to the relative susceptibility value rather than
absolute values in different models, unless we need further study such as risk assessment. It is realized that the
performance of the model is, to a great extent, determined by the effect of its classification. What's more,
comprehensive consideration of endogenic and exogenic geological processes in susceptibility assessment has
better expected results. Last but not least, under the engineering geological environment with acceptable difference,
it has advantages of practical significance to regard the administrative region as a research area for policy making.
because different regions have different status constraints in population quality and economy. In short, an effort has
been made to develop a cost- and time-efficient debris flow susceptibility assessment with an acceptable degree of
accuracy for regional-scale planning and contribute to making hazard, susceptibility and risk maps more accessible
to individuals and local authorities. The evolution of GIS-based methods and modern data availability especially
through online databases significantly contribute towards this aim. However, a challenge remains in producing





results with meaningful accuracy for the scale of planning, using available resources. Previous studies, as well as
the present work, highlight that the effectiveness of the final map depends on the quality of input data. Comparison
with a very high-resolution LIDAR-derived DEM indicated that the spatial accuracy of the DEM varies between
different landforms (lakes, river channels, riverbeds, floodplains etc.) and the areas of greatest errors are
predominantly confined to valley floors .However, with overall RMS error of 8.15 m, the DEM meets the
internationally accepted accuracy standards as set out by US Geological Survey (USGS 1997) and is of sufficient
quality for regional-scale studies such as the present one. Updating and improving existing debris flow catalogues
and inventories are crucial for the development of reliable susceptibility and risk assessment methods.
**Acknowledgements**
This research was financially supported by the Key Project of NSFC-Yunnan Joint Fund (Grant no. U1702241) and
the National Key Research and Development Plan (Grant No. 2018YFC1505301). The authors would like to thank
Yuchao Li, Zhihai Li, Jiejie Shen, Feifan Gu et al. for their contributions to the collection of field data, and the editor and
anonymous reviewers for their comments and suggestions which helped a lot in making this paper better.



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
