# Peer review of "GIS-models with fuzzy logic for Susceptibility Maps of debris flow using multiple types of parameters: A Case"

_Natural Hazards and Earth System Sciences, 2021_

## Referee Comment (RC2)

[referee-annotated manuscript omitted]

---

## Author Comment (AC1)

**Reviewer #1's Comments**

**1. General comments**

The paper entitled "Regional-scale GIS-models with fuzzy logic for Susceptibility Maps of debris flow: A Case Study in Pinggu District of Beijing, China", focused on the debris flow susceptibility map computation of a series of drainage basins of the Pinggu District of Beijing. The authors proposed a methodology based on GIS-models, combining diverse methods: grey relational method, data-driven and fuzzy logic methods. The manuscript deals with the application of susceptibility analysis on debris flow. The topic is interesting and is suitable for the journal. The model used in the manuscript not only considers the scientificity and accuracy, but also considers the application in engineering practice. I think the article can be acceptable after some revisions are made.

*Response:* Thank you very much for your valuable and constructive comments on this manuscript. Your comments are very helpful for us to improve the manuscript. Based on your comments, we have carefully revised the relevant content of the manuscript. Please see the specific responses below for more details.

**2. Specific comments**

*Comment 1:* In ArcGIS, the watershed algorithm is to obtain the sub watershed units of the whole Pinggu region. How can the author select these specific watersheds in the article? How are other unqualified units excluded?

*Response:* Thank you for your professional comments. According to your suggestions, we screened the watersheds, which is one of the features. We first removed the flat areas from the study area and then divided the remaining watersheds using the hydrology module of ArcGIS. The specific process is divided into the following steps: (1) Filling the initial digital elevation model to eliminate the common errors caused by the resolution and rounding of the data. (2) Encoding the outflow direction of each pixel in the grid based on an 8-direction algorithm. (3) Calculating accumulated flow as the accumulated weight of all cells flowing into each downslope cell in the output raster. (4) Applying a threshold to the results obtained by the flow accumulation tool based on a condition function and describing the drainage network of the study area. (5) Extracting the basic drainage basins unit of the study area, that is, the basic unit for susceptibility assessment. The fourth of five steps, threshold determination is a factor of subjective human choice, and my current research involves how to choose this parameter objectively.

In our research, typical valley debris flows are the major research object. Therefore, as shown in the figure below, A has typical watershed characteristics, but B and C do not. There is another advantage of determining the length of the main ditch in the watershed parameter characteristics. For watersheds without obvious watershed characteristics, it is difficult to determine their length from the picture. Similarly, the calculation of drainage density is very difficult.

[Figure]

***Comment 2***: How to explain the similarities and differences between models R6-R17?

***Response:*** Thank you for your professional comments. Table 5 is one of our most important
findings. It is our illustration of the results in terms of the emphasis on non-linear combinations
of factors rather than simple linear superposition. In order to characterize the development of
debris flows in an area, information mining is an effective means. The similarity of the models
is that the individual R6-R17 models are all calculated using fuzzy logistic operations and
perform well in the AUC test. The differences are that the data are pre-processed differently
when using this algorithm. However, four of the models performed above 94, mainly due to the
small number of watersheds in the study area and the small number of mudslides that could be
used for the test.

***Comment 3***: The introduction needs a section concerning susceptibility methods.

***Response:*** Thank you for your professional comments. According to your suggestions, we have
read more literature to add to the relevant research developments.

[revised manuscript text omitted]

***Comment 4***: The Results and Discussion needs to be more detailed and organized.

***Response:*** Thank you for your professional comments. According to your suggestions, we will
carefully and exhaustively sort out the ideas in the article. Then the discussion and conclusion
will be reorganized to make this section more logical and systematic as well as readable.

[revised manuscript text omitted]

**Comment 5:** Line 26 by the results → by results

*Response:* Thank you for your professional comments. According to your suggestion, we will replace the expression.

**Comment 6**: Line 26    validated by the other two → validated by two other

*Response:* Thank you for your professional comments. According to your suggestion, we will replace the expression.

**Comment 7**: Line 27    the method to → a method to

*Response:* Thank you for your professional comments. According to your suggestion, we will replace the expression.

**Comment 8**: Line 47    significance to establishing → significance to establish

*Response:* Thank you for your professional comments. According to your suggestion, we will correct the expression.

**Comment 9**: Line 76    disaster chain and that the geomorphic → disaster chain and the geomorphic

*Response:* Thank you for your professional comments. According to your suggestion, we will replace the expression.

**Comment 10**: Line 76    rather than simple data fitting → rather than simply data fitting

*Response:* Thank you for your professional comments. According to your suggestion, we will correct the expression.

**Comment 11**: Line 80    account for → accounts for

*Response:* Thank you for your professional comments. According to your suggestion, we will correct the expression.

**Comment 12**: Line 90    1.    Data and Methodology → 3 Data and Methodology

*Response:* Thank you for your professional comments. According to your suggestion, we will correct the number.

**Comment 13**: Line 99    watershed characteristics factors → watershed characteristic factors

*Response:* Thank you for your professional comments. According to your suggestion, we will correct the mistakes.

**Comment 14**: Line 103    our primary assumption here are → our primary assumptions here are

*Response:* Thank you for your professional comments. According to your suggestion, we will correct the expression.

**Comment 15**: Line 103    First → Firstly

*Response:* Thank you for your professional comments. According to your suggestion, we will replace the expression.

**Comment 16**: Line 105    Second → Secondly

*Response:* Thank you for your professional comments. According to your suggestion, we will
replace the expression.

*Comment 17:* Line 114   by professional team → by professional teams

*Response:* Thank you for your professional comments. According to your suggestion, we will
replace the expression.

*Comment 18:* Line 138   factors (Type B) factors → factors (Type B)

*Response:* Thank you for your professional comments. According to your suggestion, we will
correct the mistake..

*Comment 19:* Line 159   a effective method → an effective method

*Response:* Thank you for your professional comments. According to your suggestion, we will
correct the mistake.

*Comment 20:* Line 169   3.4 fuzzy memberships → 3.4 Fuzzy memberships

*Response:* Thank you for your professional comments. According to your suggestion, we will
correct the mistake.

*Comment 21:* Line 217   can be used to derived their fuzzy → can be used to derive their fuzzy

*Response:* Thank you for your professional comments. According to your suggestion, we will
correct the mistake.

*Comment 22:* Line 238   order to use properly → order to use it properly

*Response:* Thank you for your professional comments. According to your suggestion, we will
correct the mistake.

*Comment 23:* Line 247   Compared with other four fuzzy operator → Compared with other four
fuzzy operators

*Response:* Thank you for your professional comments. According to your suggestion, we will
correct the mistake.

*Comment 24:* Line 247   Fuzzy Gamma (Eq.6) → Eq.5

*Response:* Thank you for your professional comments. According to your suggestion, we will
correct the mistake.

*Comment 25:* Line 261   seventeen results were compared (Table.6) → Table.5

*Response:* Thank you for your professional comments. According to your suggestion, we will
correct the mistake.

*Comment 26:* Line 278   the results is not comprehensive → the results are not as comprehensive

*Response:* Thank you for your professional comments. According to your suggestion, we will
correct the mistake.

**Comment 27**: Line 282    there are total 135 basin → there are total 135 basins

*Response:* Thank you for your professional comments. According to your suggestion, we will
correct the mistake.

**Comment 28**: Line 306    uncertain factor compared with factors compared → uncertain factors compared

*Response:* Thank you for your professional comments. According to your suggestion, we will
correct the mistake.

**Comment 29**: Line 309    bedrock fracture flow; and root strength → bedrock fracture flow, and root strength

*Response:* Thank you for your professional comments. According to your suggestion, we will
correct the mistake.

**Comment 30**: Line 334    in which all factors as a single → in which all factors are considered as a single

*Response:* Thank you for your professional comments. According to your suggestion, we will
replace the expression.

**Comment 31**: Line 362    nonlinear methods is consistent → nonlinear method is consistent

*Response:* Thank you for your professional comments. According to your suggestion, we will
correct the mistake.

**Comment 32**: Line 377    clear and the data easy to obtain → clear and the data is easy to obtain

*Response:* Thank you for your professional comments. According to your suggestion, we will
correct the mistake.

We thank you very much for not lowering your evaluation because of our poor English. The
recognition of the value of our research is a great encouragement to us. We have tried our best to
improve the manuscript and made changes in the manuscript. We appreciate for your warm work
earnestly, and hope that the result will meet with approval. As the manuscript has undergone
several previous revisions, there are many errors in detail. We apologize for the very bad effect on
your reading experience. Once again, thank you very much for your comments and suggestions!
Please feel free to contact me, if any further changes are required. We look forward to hearing
from you.

Yours sincerely,

Jianping Chen, Ph.D.

College of Construction Engineering, Jilin University

938 Ximinzhu Road, Changchun 130026, China

Phone number: +86 13843047952

Email address: chenjp@jlu.edu.cn

---

## Author Comment (AC2)

**Reviewer #2's Comments**

**1. General comments**

I have only partially revised the manuscript "GIS-models with fuzzy logic for Susceptibility Maps of debris flow using multiple types of parameters: A Case Study in Pinggu District of Beijing, China". The manuscript deals with the application of susceptibility analysis on debris flow and could be interesting for the journal. Unfortunately, the manuscript is not written in a good English and many statements and descriptions are very difficult to understand. I revised only up to line 203 (3.4.2 Data-driven method in susceptibility modelling). I recommend the authors to submit a revised version of the manuscript after the revision of an English-speaking person. Few comments are throughout the text.

*Response:* Thank you very much for your valuable and constructive comments on this manuscript. Your comments are very helpful for us to improve the manuscript. Thank you for your suggestions on the language. These suggestions were of great help and improved the quality of our manuscript. According to your suggestions, we will send our manuscript to professional language embellishment agency and foreign students who are English-speaking person. Hope the final revision can meet your requirements. In the following, we will reply to and explain the language comments one by one to clarify our intended meaning. Please see the specific responses below for more details.

**2. Specific comments**

*Comment 1:* Line 54, the sentence "in the early days, the susceptibility assessment of debris flows was mainly qualitative research" is not completely true.

*Response:* Thank you for your professional comments. We apologize for the misunderstanding caused by our expression. We would like to say that before 1970, the limitations of remote sensing and computer technology caused more studies to be expressed without a very precise quantification. Based on your comments and a review of the relevant literature, we think it is more appropriate to remove this ambiguous expression.

*Comment 2:* Line 62, the sentence "Surely, they are also wasteful and unnecessary" has English problem

*Response:* Thank you for your professional comments. We apologize for the misunderstanding caused by our expression. We have reread this paragraph and consider it redundant and ambiguous. The sentence should be deleted.

*Comment 3*: Line 62, what is 3S?

*Response:* Thank you for your professional comments. 3S is mean 3S technology, which is Remote sensing, Geography information systems, Global positioning systems. We apologize for the use of abbreviations without explanation.

*Comment 4*: Line 66, the sentence "While due to the nonlinearity of debris flow system and the openness and complexity of geological environment, we realize that it is chaotic, with many factors affecting the system." need to be revised.

*Response:* Thank you for your professional comments. We reorganized the intention and made it clearer. It is revised below. "As research progresses, debris flows are increasingly seen as an open system. There are many factors influencing the system and the combination of factors is non-linear and the interactions are chaotic."

*Comment 5*: Line 73-76, the sentence "According to the summary above, the primary object of my present study is to explore a geographic information system (GIS)-based quantitative model based on expert experience and field investigation. And the model is consistent with the system characteristics of debris flow gully and can also indicate the characteristics of disaster chain and that the geomorphic evolution of basin rather than simple data fitting(Porwal et al. 2006)." has English problem, it is not clear and correct.

*Response:* Thank you for your professional comments. We apologize for any confusion caused by the lack of English expression skills. We have rewritten the sentence below. "The main objective of this paper is to propose a quantitative geographic information system (GIS)-based model. The results of expert experience scoring and site surveys are used as guidance and reference in the modelling process. We have tried to apply methods that can indicate the non-linearity of the debris flow system. Finally, the modelling process should respect the laws of geomorphological evolution and the geological basis. Otherwise, the result will tend to be simply data fitting (Porwal et al. 2006)."

*Comment 6*: Line 79, terrain should be replaced by elevation

*Response:* Thank you for your professional comments. We have replaced the word.

*Comment 7*: Line 84, the sentence "political factors must be taken into account" is not clear

*Response:* Thank you for your professional comments. Different administrative regions often have different financial incomes. The situation will lead to different standards and economic investments in the prevention and treatment of geological hazards. Therefore, different decisions will be made for hazards of the same level. This is what we mean by "political factors".

**Comment 8**: Line 87, explain the meaning of the sentence "precision of the base map and the size of the study area".

*Response:* Thank you for your professional comments. Base map mainly refers to geological map and digital elevation map (DEM) in this paper. The geological map is 1: 50 000 and the accuracy of dem is 30 m. We think the above precision is suitable for the study area. In other words, it is not appropriate to use the above-mentioned precision map to study global scales.

**Comment 9**: Line 91, the sentence "drainage basins unit", explain what they area

*Response:* Thank you for your professional comments. The "drainage basins unit" are showed in Fig.4 line 128.

**Comment 10**: Line 95, explain the sentence "obvious watershed characteristics"

*Response:* Thank you for your professional comments. In our research, typical valley debris flows are the major research object. Therefore, as shown in the figure below, A has typical watershed characteristics, but B and C do not. There is another advantage of determining the length of the main ditch in the watershed parameter characteristics. For watersheds without obvious watershed characteristics, it is difficult to determine their length from the picture. Similarly, the calculation of drainage density is very difficult.

[Figure]

**Comment 11:** Line 98-99, the sentence "it is scientific to make full use of qualitative understanding to determine the weight of the parameters of watershed characteristics factors" is not clear.

*Response:* Thank you for your professional comments. We have reorganized our language: field inspection is generally required in geological hazard surveys. If the data from the field inspection is applied to the model, it can help the model building and reduce the time for model training. The weights derived from the grey relational analysis method used in the following section (in section 3.4.1) are based on the data from the field inspection.

**Comment 12:** Line 102, explain better the workflow

*Response:* Thank you for your professional comments. First, a DEM map of the Pinggu area was downloaded. Then, the basin units are then generated from the DEM map using the ArcHydro tool. The derived results were analyzed and units that did not fit the characteristics of the watershed were removed. During the analysis, the field survey data and Google images were referenced. After that, the controlling and triggering factors for the remaining 135 catchments were counted. For the fuzzy memberships, watershed characteristic parameters were determined by grey correlation and the geological and geomorphological factors were determined by the frequency ratio (FR) method and the cosine amplitude method. Finally, the individual layers were overlayed by fuzzy logic operations to obtain the final assessment map. As there were different combinations of factors, 17 results were derived. In order to compare advantages and disadvantages of these results, three indexes, AUC, AR and RR, were used to evaluate the models.

**Comment 13:** Line 104-105, this is also a local property

*Response:* Thank you for your professional comments. This statement was made to emphasize the importance of micro-landscapes in the evaluation, which is why we included the parameter roughness in the model.

**Comment 14:** Line 107-109, the sentence "Finally, the model should also need to integrate the system characteristics of debris flow disaster, the future development trend of climate change, and the social demand under the theoretical background of the new era to carry out reasonable modeling" has English problem

*Response:* Thank you for your professional comments. The sentence has been revised. "Finally, the model is expected to reflect the system characteristics, the trend of climate change, and the social demand."

*Comment 15*: Line 111, The workflow should be explained.

*Response:* Thank you for your professional comments. I have replied to this comment. Please refer to *Comment 13* above for details.

*Comment 16*: Line 112, explain better how did you completed the debris flows inventory

*Response:* Thank you for your professional comments. All the cataloguing process is carried out on the ArcGIS software. The specific process is divided into the following steps: (1) Filling the initial digital elevation model to eliminate the common errors caused by the resolution and rounding of the data. (2) Encoding the outflow direction of each pixel in the grid based on an

8-direction algorithm. (3) Calculating accumulated flow as the accumulated weight of all cells flowing into each downslope cell in the output raster. (4) Applying a threshold to the results obtained by the flow accumulation tool based on a condition function and describing the drainage network of the study area. (5) Extracting the basic drainage basins unit of the study area, that is, the basic unit for susceptibility assessment. The fourth of five steps, threshold determination is a factor of subjective human choice, and my current research involves how to choose this parameter objectively.

*Comment 17*: Line 119, what is difference with the slope unit.

*Response:* Thank you for your professional comments. In general, the main difference is the way in which they are defined. A slope unit is a basic closed unit enclosed by a ridge and valley line.

Basin units, on the other hand, often consist of at least two slope units. This is shown in the figure below.

[Figure]

*Comment 18*: Line 120, the phrase "irregular areas". What do you mean? It is not clear why you have selected only 135 basins.

*Response:* Thank you for your professional comments. irregular areas refers to areas which are not basin units automatically generated by using ArcHydro tool, such as slope unit in *Comment 18.*

We admit that there is a certain subjective component (extent depending on the accuracy of the

DEM), but it is proven to be an attempt to improve the accuracy of the model. When deleted and merged, there are 135 basins left.

**Comment 19:** Line 133, facros should be replaced by factors.

*Response:* Thank you for your professional comments. We have replaced the wrong word.

**Comment 20:** Line 135, the phrase "in this paper" should be deleted.

*Response:* Thank you for your professional comments. It has been deleted.

**Comment 21:** Line 138, the sentence "is bounded by the watershed". What do you mean?

*Response:* Thank you for your professional comments. The statistics for these factors are based on the watershed as a basic unit and the parameters change as the delineated watershed changes. Geological factors, however, are not bound by geological boundary lines. For example, the same stratigraphic lithology can span several watersheds.

**Comment 22:** Line 143-144, we indirectly consider the influence of natural loose material source by evaluating geological conditions, but cannot consider the impact of human activities. It is not clear what is the relationship between the two factors

*Response:* Thank you for your professional comments. The sources of debris flow in the study area include both naturally occurring and anthropogenic sources (road construction, mining). Natural sources can be evaluated indirectly by relevant factors (geological and geomorphological conditions), but the intensity of anthropogenic sources cannot be predicted. Moreover, the thickness cannot be clearly counted on remote sensing images. Therefore, the evaluation factor can indirectly consider the influence of natural loose material source, but not human-generated loose source (slag, gravel soil, etc.)

**Comment 23:** Line 149 in Table 1, the phrase "derived from DEM". Automatically?

*Response:* Thank you for your professional comments. It is not derived automatically. Firstly, we first determine the scope of the basin according to DEM. When the scope is determined, it can be directly in ArcGIS 10 2 calculate and count the projected area value of each watershed. The rest of the factors are the same steps

**Comment 24:** Line 149 in Table 1, the word "numerical" should be deleted.

*Response:* Thank you for your professional comments. It has been deleted.

**Comment 25:** Line 149 in Table 1, the sentence "higher frequency of slope failures" is not always true.

*Response:* Thank you for your professional comments. We understand what you arguement, that this is not a linear increase. what we are describing is that all other conditions are constant and only this one variable is present. In terms of mechanics , the greater the slope, the greater the downward component of gravity, and the more likely it is to slide. We will try other expressions to prevent this ambiguity.

[Figure]

**Comment 26:** Line 156, "curve length" is not clear. Why curve?

*Response:* Thank you for your professional comments. This is a mathematical concept. As shown in Fig. 5, relative to the linear connection between two points ($A_7$), the connection line is called curve line in this paper. And its length is called curve length.

**Comment 27:** Line 159-161, the sentence "Fuzzy set theory proposed by Zadeh (1965) is a effective method to express the concept of partial set membership degree. This concept is different from the classical binary (two-valued) logic by using fuzzy descriptions such as low, moderate, high, steep, favourable and close to (Kritikos and Davies 2015)." Should be rephrased.

*Response:* Thank you for your professional comments. The sentence has been rephased. "Fuzzy set theory is proposed by Zadeh (1965). It is an efficient way of expressing the concept of partial set membership degree. This concept differs from classical binary(0-1 value) logic. More words with a transitional fuzzy descriptions (such as low, medium, and high) are used (Kritikos and Davies 2015). This fuzzy expression is particularly applicable to geological hazard classification."

**Comment 28:** Line 191 in table 2, why only same basins are shown in the table.

*Response:* Thank you for your professional comments. "gully" represents "the name of the gully", "score" represents "the score of the gully". We have modified the format to remove the ambiguity.

**Comment 29**: Line 191 in table 2, where this score comes from?

*Response:* Thank you for your professional comments. According to the "Specifications for Geological Investigation of Debris Flows Stabilization (DZ/T0220-2006) (2006)" published by the China Ministry of Lands and Resources. It is an industry standard that we need to follow for field surveys. Likewise, if in another country, people could use their local standards. This is also the flexibility of the model

[Figure]

**Comment 30**: Line 196-197, the sentence "it can be seen from the results that the occurrence of debris flow is highly correlated with basin volume, basin area and main gully bending coefficient with fuzzy membership above 0.7 in Beijing area." How do you explain this behaviour.

*Response:* Thank you for your professional comments. This is a regional attribute and regular characteristics of debris flow development in the study area. Debris flows occur mostly during the rainy season (June to August). Moreover, the study area is characterized by short duration heavy rainfall and the distribution of rainfall is not significantly different across the study area. The source of the loose material therefore becomes the dominant factor. And the three factors mentioned above are highly correlated with total physical sources. Both basin area and basin volume determine the upper limit of the maximum source, while the bending factor directly influences the replenishment of loose sources along the debris flow ditch.

[Figure]

Monthly rainfall in Pinggu district, Beijing, 2018

**Comment 31:** Line 197-202, the English expressions "In the case of sufficient rainfall, the basin directly determines the total amount of catchment, and the bending coefficient reflects the replenishment of the source along the river. The basin volume is closely related to the number of supplementary sources. Therefore, it is necessary to do well in rainfall monitoring and early warning in large watersheds, check for loose matter accumulation in river basins before rainy season, and pay attention to slope protection of basin with large volume potential energy for the purpose of disaster prevention and reduction" should be revised.

**Response:** Thank you for your professional comments. The sentence has been revised. "Rainfall in the study area is abundant to induce the debris flow. Loose source and sinks the total volume of catchment become more important. The watershed area determines the total volume of catchment. For the same rainfall, generally, the larger the area, the larger the catchment is. The bending coefficient reflects the replenishment sources along the channel. The greater the coefficient, the slower the flow is. Then loose source along the channel has more time to replenish. Basin volume characterizes the maximum amount of loose material that can be supplied. These three features reflect the development characteristics of debris flow in the study area. It also provides ideas for disaster prevention and mitigation.

**Comment 32:** Line 204, the expression "landslide is one of the main fixed sources of debris flow" is not clear.

**Response:** Thank you for your professional comments. Excluding human activities, such as mining, construction, etc., loose material produced by natural geological processes is the primary source of debris flow formation. Great debris flows may result from numerous, small slope failures that subsequently coalesce (Fairchild 1987; Roeloffs 1996), from flow enlargement due to incorporation of bed and bank debris (Bovis and Dagg 1992; Pierson et al. 1990), or from large, individual landslides that mobilize partially or almost totally (Iverson et al. 1997; Vallance and Scott 1997). Debris flows may also scour steep channels to bedrock and accelerate sediment delivery to downstream, lower-gradient channels. The spatial and temporal distribution of shallow landslides are important controls on landscape evolution and a major component of both natural and management-related disturbance regimes in mountain drainage basins (Benda 1987; Crozier et al. 1990; Dietrich et al. 1986; Tsukamoto et al. 1982).

Cited Reference:

Fairchild LH (1987) The importance of lahar initiation processes Reviews in Engineering Geology 7:51-62 doi:10.1130/REG7-p51

Roeloffs E (1996) Poroelastic techniques in the study of earthquake-related hydrologic phenomena 38:135-195 doi:10.1016/S0065-2687(08)60270-8

Bovis M, Dagg B (1992) Debris flow triggering by impulsive loading - mechanical modeling and case-studies Canadian Geotechnical Journal 29:345-352 doi:10.1139/t

Iverson RM (1997) The physics of debris flows Reviews of Geophysics 35:245-296. doi:10.1029/97RG00426

Vallance JW, Scott KM (1997) The Osceola mudflow from mount rainier: Sedimentology and hazard implications of a huge clay-rich debris flow Geological Society of America Bulletin 109:143-163 doi:10.1130/0016-7606(1997)109<0143:TOMFMR>2.3.CO;2

Pierson TC, Janda RJ, Thouret J-C, Borrero CA (1990) Perturbation and melting of snow and ice by the 13 November 1985 eruption of Nevado del Ruiz, Colombia, and consequent mobilization, flow and deposition of lahars Journal of Volcanology and Geothermal Research 41:17-66 doi:10.1016/0377-0273(90)90082-q

Thank you for your professional comments. We apologize for the bad reading experience caused by our poor English. We also hope that language issues will not become a barrier to scientific communication and that you will Reconsidering our research beyond the language issue. We will try our best to improve the manuscript and make changes in the manuscript. We appreciate for Editors/Reviewer's warm work earnestly, and hope that the revision will meet with approval. Once again, thank you very much for your comments and suggestions! Please feel free to contact me, if any further changes are required. We look forward to hearing from you.

Yours sincerely,

Qing Wang, Ph.D.

College of Construction Engineering, Jilin University

Ximinzhu Road, Changchun 130026, China

Phone number: +86 13843047952

E-mail: wangqing@jlu.edu.cn

---

## Author Response (AR1)

**Editor's or Referees 's Comments**

**1. General comments**

Thank you for the submission of your manuscript "GIS-models with fuzzy logic for Susceptibility Maps of debris flow using multiple types of parameters: A Case Study in Pinggu District of Beijing, China". As you know, two reviewers have now provided detailed reviews, which you have replied to. One reviewer recommended minor to medium revisions, the other one to reject the manuscript. I believe that your manuscript needs tremendous improvement to bring it up to an international level before it can be further reviewed. The main issues at this point, which will require a major rewrite and revision are as follows:

*Response:* Thank you very much for your valuable and constructive comments on this manuscript. Your comments are very helpful for us to improve the manuscript. In the following, we will reply to explain the comments one by one to clarify our intended meaning. Please see the specific responses below for more details.

**2. Specific comments**

**Comment 1:** (a) NOVELTY OF YOUR STUDY. Your research on the application of susceptibility analysis on debris flow is a more or less interesting case study, but you do not tell us what is novel. This needs to be done both at the beginning so we understand, but also in discussion, telling us 'why should someone outside of your study area be interested in the results'. If you were to explain the results of your case study to someone in another country, what would they gain from your case study? Do they learn from your methodology and what you encountered when applying it? What is novel and what might they learn?

*Response:* Thank you very much for your valuable and constructive comments on this manuscript. Your comments are very helpful for us to improve the manuscript. We summarize the novelty of our paper as follows:

1. We summarize the commonly used mudflow evaluation indicators, define and explain the role of each indicator in detail. On this basis, a new factor was proposed which contributes up to 0.79, indicating that this factor should be given attention. The factor evaluates the debris flow from an energy perspective.

2. 17 models were derived through a scientific method of controlling variables in a case study. By comparing the results of the models, it is found that grouping the influencing factors helps to improve the accuracy of the models, i.e., those with similar intrinsic properties should be overlaid into one group. Then the result of these group will be calculated with each other. In contrast, previous studies merely superimposed the factor layers individually. If the study

object is linear, then the results are consistent. If the study object is nonlinear, then the results tend to be different. This research idea verifies that the debris flow system has nonlinear characteristics.

3. AUC is an effective index for evaluating models, but there are certain limitations. For basins where debris flows have occurred, we can define them as positive; but for basins that are not currently occurring, we can't define them as negative theoretically. In other words, the applicability of the method gradually increases when there is abundant data in the area in the past. But when the region is data-poor, many classifications are manual divisions, which are prone to bias. Therefore, two other indicators are proposed in our paper.

4. Only one of the 17 models proposed in this paper has an AUC below 0.6, indicating that the modeling logic is reasonable. Coupled with the simplicity of the method used, the small demand for data, and the clear meaning of the factors, these advantages ensure that the model is transferable to other regions.

**Comment 2**: (b) BROADER CONTEXT OF YOUR STUDY. You do not relate your work to the broader literature of what others have done. We need to understand this broader context and what others have done.

*Response:* Thank you for your professional comments. We have read the relevant literature and added the relevant content to the ***introduction*** section.

**Comment 3**: (c) ENGLISH. Although your manuscript will undergo a copy editing at the final stage, there are sentences in your manuscript which one cannot follow due to the issues of English. I am not saying the English must be grammatically perfect, but at least to a level that the reviewers (and myself) can understand what is being said scientifically.

*Response:* Thank you for your professional comments. We apologize for the misunderstanding caused by our expression. It is supposed that the language issue you mentioned is very pertinent. We have read our manuscript carefully again and revised the language significantly as well as the structure.

**Comment 4**: You need to do an extensive revision of your manuscript before resubmit it.

*Response:* Thank you for your professional comments. Last time your email told me not to revise the original draft and to reply first, so I did not revise the original draft. This time, I have made a lot of revisions based on the relevant issues, and I hope it will meet your requirements and those of the reviewers. I sincerely hope that our research results can be published in your journal and

69     can be related in the academic community.

70

71        Thank you for your professional comments. We apologize for the bad reading experience
72     caused by our poor English. We also hope that language issues will not become a barrier to
73     scientific communication. We have tried our best to improve the manuscript and make changes in
74     the manuscript. We appreciate for Editors/Reviewer's warm work earnestly, and hope that the
75     revision will meet with approval. Once again, thank you very much for your comments and
76     suggestions! Please feel free to contact me, if any further changes are required. We look forward
77     to hearing from you.

78     Yours sincerely,

79     Jianping Chen, Ph.D.

80     College of Construction Engineering, Jilin University
81     938 Ximinzhu Road, Changchun 130026, China
82     Phone number: +86 13843047952
83     E-mail: chenjp@jlu.edu.cn

---

## Author Response (AR2)

**Editor's or Referees 's Comments**

**1. General comments**

The paper "GIS-models with fuzzy logic for Susceptibility Maps of debris flow using multiple types of parameters: A Case Study in Pinggu District of Beijing, China" presents a quantitative susceptibility assessment model for debris flow, which was established in the Pinggu District of Beijing. The authors claim that the method and the resulting susceptibility map are particularly helpful for decision makers in dealing with regional-scale land use planning and debris flow hazard mitigation in data scarce mountainous areas.

I agree with the authors, that such a simple but sound and sufficiently accurate method is helpful in practical terms. Still, there are aspects which need to be improved to make this manuscript interesting for the international scientific community.

*Response:* Thank you very much for your valuable and constructive comments on this manuscript. We have carefully studied your comments. Your summary of our article is very accurate and your comments are very helpful for us to improve the manuscript. In the following, we will reply to explain the comments one by one. Please see the specific responses below for more details.

**2. Specific comments**

*Comment 1*: (a) NOVELTY OF YOUR STUDY. Your presented model established in the Pinggu District of Beijing is an interesting case study, but you do not sufficiently tell us what is novel. This needs to be done both at the beginning so we understand, but also in discussion, telling us 'why should someone outside of your study area be interested in the results'. If you were to explain the results of your case study to someone in another country, what would they gain from your case study? Do they learn from your methodology and what you encountered when applying it? What is novel and what might they learn?

*Response:* Thank you very much for your valuable and constructive comments on this manuscript. Your comments are very helpful for us to improve the manuscript. We have revised our manuscript to complement our novelty in the section *Abstract* (line 27-30) *and Conclusion* (line 370-374).

*Comment 2*: (b) DECISION MAKING UNDER UNCERTAINTY. The accuracy of your models are assessed in detail, but what does this mean for decision makers. Please elaborate more about communicating uncertainty, how can decision makers deal with different levels of uncertainty and how can quantitative information about uncertainty be used in decision making processes.

*Response:* Thank you for your professional comments. We have revised the original manuscript in
the section *Results and Discussion* (line 351-357) as follows:

Finally, we should consider decision making under uncertainty, because the debris flow
phenomenon is extremely complex. The classification of geologists (high, moderate and low) is
ambiguous for decision makers. It is more beneficial for them to use mathematically rigorous
definitions. Considering that geological conditions tend to vary greatly from region to region, it is
not appropriate to define a fixed limit. the Jenks method (chosen in this paper) can be used to
classify sensitivity maps according to the characteristics of the data itself. We can also further
process the data according to the needs of decision makers, such as identifying 10% of the
watersheds in the entire region as very high risk. However, the applicability of the model to
extreme rainfall and seismic conditions is not considered.

*Comment 3*: (c) ENGLISH. Although your manuscript will undergo a copy editing at the
final stage, there are still sentences in your manuscript which one cannot follow due to the issues
of English. Further improvement of your English text is necessary.

*Response:* Thank you for your professional comments. We have read our manuscript carefully
again and revised the language. We believe the new revision is more suitable to read and
understand.

*Comment 4*: (d) CONCLUSION. Please remove the parts which are just summarising your
results from the conclusion. This conclusion part of the paper should be as concise as possible.

*Response:* Thank you for your professional comments. We have revised the original manuscript in
the section *Conclusion* and deleted some duplicate content.

Thank you for your professional comments. We have tried our best to improve the
manuscript and make changes in the manuscript. We appreciate for Editors/Reviewer's warm
work earnestly, and hope that the revision will meet with approval. Once again, thank you very
much for your comments and suggestions! Please feel free to contact me, if any further changes
are required. We look forward to hearing from you.

Yours sincerely,

Jianping Chen, Ph.D.

College of Construction Engineering, Jilin University
938 Ximinzhu Road, Changchun 130026, China

Phone number: +86 13843047952

E-mail: chenjp@jlu.edu.cn